# The Unexplored Socio-Cultural Benefits of Coffee Plants: Implications for the Sustainable Management of Ethiopia's Coffee Forests

**Bikila Jabessa Bulitta** [1] **and Lalisa A. Duguma** [2,*]

1   College of Social Sciences and Humanities, Department of Sociology, Dembi Dolo University, P.O. Box 260, Dembi Dollo, Ethiopia; BIKILA.BULITTA@GMAIL.COM
2   Landscapes Governance Theme, World Agroforestry (ICRAF), P.O. Box 30677, Nairobi, Kenya
*   Correspondence: L.A.DUGUMA@CGIAR.ORG

**Abstract:** Coffee is among the most popular commodity crops around the globe and supports the livelihoods of millions of households along its value chain. Historically, the broader understanding of the roles of coffee has been limited to its commercial value, which largely is derived from coffee, the drink. This study, using in-depth interviews and focus group discussions, explores some of the unrevealed socio-cultural services of coffee of which many people are not aware. The study was conducted in Gomma district, Jimma Zone, Oromia National Regional state, Ethiopia, where arabica coffee was first discovered in its natural habitat. Relying on a case study approach, our study uses ethnographic study methods whereby results are presented from the communities' perspectives and the subsequent discussions with the communities on how the community perspectives could help to better manage coffee ecosystems. Coffee's utilities and symbolic functions are numerous—food and drink, commodity crop, religious object, communication medium, heritage and inheritance. Most of the socio-cultural services are not widely known, and hence are not part of the benefits accounting of coffee systems. Understanding and including such socio-cultural benefits into the wider benefits of coffee systems could help in promoting improved management of the Ethiopian coffee forests that are the natural gene pools of this highly valuable crop.

**Keywords:** socio-cultural benefits; Gomma; symbolic; utility; coffee forest; society

## 1. Introduction

If there is one crop that is widely known globally for its use in a large number of societies and households, it is coffee. It is considered one of the most popular drinks in the world and is enjoyed by over one billion of people, according to industry estimates. Globally, every single day, around 2.25 billion cups of coffee are consumed [1]. The plant from which coffee comes is almost entirely grown in developing countries, the source of 90% of the production. Although a number of species of the genus *Coffea* produce coffee, the most popular are *Coffea arabica* L. and *Coffea canephora* Pierre ex A. Froehner (syn. *Coffea robusta*). The former generates about 70% of the global production, while the latter produces the remaining 30% [2]. *Coffee arabica* L. is believed to have its origin in Ethiopia [3,4], with many studies confirming this through genetic mapping [5]. The montane highlands of Ethiopia are the only known origin of arabica coffee and harbor the genetic diversity pool [5,6]. Owing to its wide genetic diversity in this region [7], the latest studies are even discovering naturally decaffeinated coffee [8]. The conservation of coffee forests is crucial for safeguarding this natural genetic pool and also to secure subsistence coffee production in the future [9]. It is this global recognition of its genetic and economic importance that led to the development of a UNESCO heritage site named Yayu Biosphere Reserve in Yayu area, southwest Ethiopia.

In 2018, the gross production value of coffee was estimated at USD 21.53 billion, almost 2.5-fold greater than that of cocoa beans according to FAOSTAT (Online). This makes it

one of the most important tree commodities globally. In Ethiopia, coffee, along its value chain, supports the livelihoods of over 15 million people [10]. This has made it one of the most important economic pillars of the country for decades. Being one of Ethiopia's largest export crops, it also supports the livelihood of millions of people outside the country who are employed along the value chains of this cash crop.

Despite coffee's wide importance, the natural habitats where its gene pools exist are under threat from the expansion of other land uses that generate quick economic returns such as crop farms and plantation coffee. Wild coffee forests in southwest Ethiopia shrank by almost 36% between 1973 and 2010 [11,12]. The underlying drivers of this loss include human activities such as the conversion of coffee forests to low-shade coffee systems to boost productivity and production to reap more from coffee farming; this involves the removal of important shade trees, which gradually changes the vegetation canopy configuration below. Another major human activity threat is the marked expansion of coffee plantations (as opposed to forest and semi-forest coffee systems). Climate change and variability is another driver [13–16], and studies are already suggesting that if no action is taken, coffee may gradually shift its growing region, making some of the current coffee growing areas unfit for coffee production. These findings are worrying, especially for communities such as those in Gomma, the socio-cultural and livelihood practices of which are strongly anchored to coffee and the forests that harbor it. The wild variant of arabica coffee is among the most threatened species close to extinction [17].

For many decades, due to its commodity nature, the worldview on coffee has been dominated by its economic and monetizable livelihood benefits. From an outsider point of view, this is largely justified. However, this way of looking at coffee undermines the wider values that the crop has, particularly the merits of the socio-ecological systems in which it is produced. Exploring and expanding our understanding of the broader values of the crop helps to implement the required conservation measures so that it thrives, and the communities and biodiversity that depend on coffee also benefit from the ecosystem services that the coffee forests generate. Incentives for local actors to conserve such globally important ecosystems depend upon how much we understand and value the broader benefits they generate. To achieve better management of this vital ecosystem, communities need to be compensated in proportion to their worth.

In this study, we adopt a sociological research approach, largely using ethnographic study principles, and we apply it to communities who have traditionally raised coffee for generations as part of their life. An in-depth assessment was conducted in Gomma district, Jimma Zone Ethiopia to uncover the wider benefits of this commodity tree crop. The paper is structured as follows: following this introduction, the first part presents a conceptual basis for understanding the broader worldview of the study community, particularly related to coffee; this is followed by a section presenting the methods for data collection. The next section presents community accounts of the socio-cultural and religious benefits of coffee. Lastly, we present the implications of these largely little known and unfamiliar benefits for the sustainable management of coffee forests in southwest Ethiopia. The authors would like to note that the sentiments expressed in this study originate from and belong to the communities and remain their heritage. For this reason, most of what the community expressed is presented in quotes in their own language in the way it is spelled by the community.

## 2. Conceptual Framing: Symbolic Anthropology and Utility Concepts

A symbol, as defined in Merriam Webster Dictionary, is "something that stands for or suggests something else by reason of relationship, association, convention, or accidental resemblance". As argued by Hoskins [18], the representation of a given symbol varies depending on the cultural contexts of a society. What a given symbol represents in one culture may be different in another culture. Thus, the use of symbols becomes a more context-specific issue than generalizable facts or representations. Symbolic anthropology concerns the interpretive quest, in which one has to understand the deeper meanings of

things and symbols within a particular socio-cultural setting [19]. Symbolic and interpretive anthropology strives to understand how people, communities, or societies give meanings to the things around them (often referred to as reality) and how this reality is represented by using symbols. Plant symbolism (use of plants representing some hidden interpretations) is a common practice in numerous countries [20–24].

For this study, we chose to use the anthropological aspect of understanding the significance of coffee in Gomma area, Ethiopia. The basis for choosing this approach beyond what is described in the paragraph above is that there is an intense need for conservationists, land managers and users, and communities to understand how such plants that are grown by almost every household in the area are seen and managed. This quest of the authors of this study is very well reflected in how Keesing et al. [19] defines the goal of anthropological studies: *"Anthropology is an exploration, an excavation, of the cumulated, embodied symbols of other peoples, a search for meanings, for hidden connections, for deeper saliences than those presented by the surface evidence of ethnography. Taking cultures as texts, symbolic anthropology seeks to read them deeply: to find . . . the 'reverberations' of a culture in ritual, in metaphor, in the meanings of everyday life"*. The most representative source of information to understand the meanings and representations of things is the community that dwells in the context and interacts with the subjects/things on a daily basis. This is crucial because meanings are complex constructs that need to be understood at a deeper level. The meanings also need validation as to whether they can be generalizable within a given context, e.g., society, time, or place.

We therefore use the symbolism aspects of coffee with the utilities context that the crop provides for the community. Utilities in this case refers to the wants and needs that the crop satisfies for the community, similar to the economic definition of the term. We believe that such in-depth insights into the broader values of the species could improve its management in its native home ranges, such as Gomma. We framed the following schematic (Figure 1) to visualize the symbolism and utility aspects of coffee in the Gomma area, based on the authors' prior experiences which motivated them to conduct the in-depth study reported here. The scheme presents the potential worldview of the Gomma Oromo communities regarding the coffee plant.

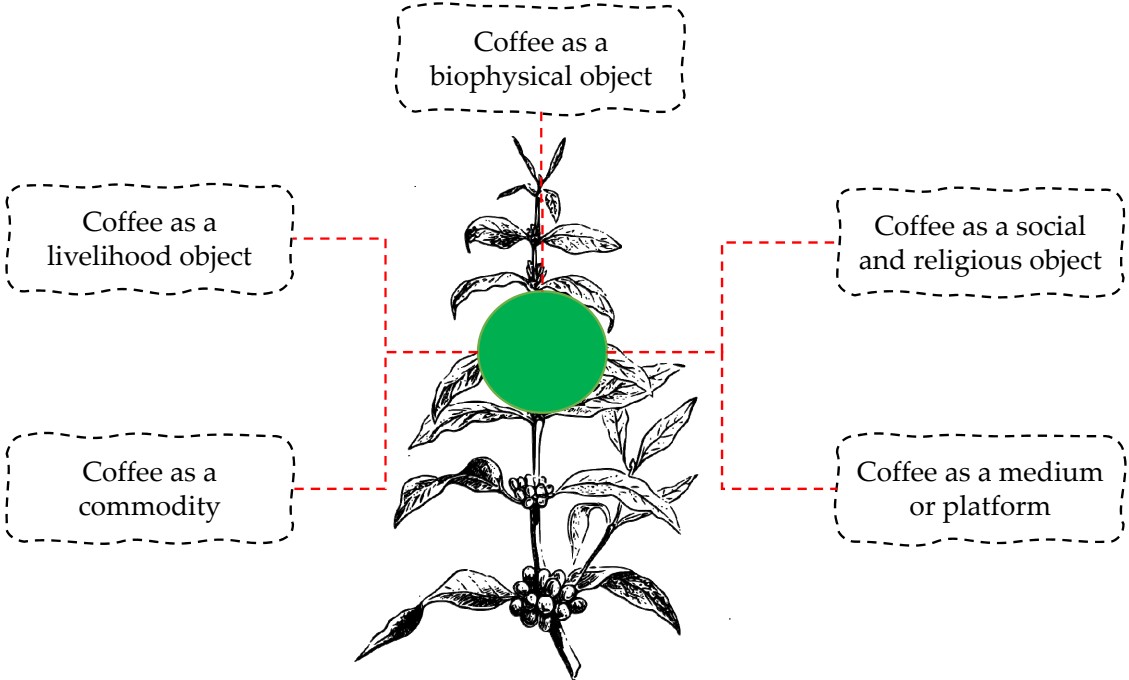

**Figure 1.** Schematic showing symbolism and utilities of coffee in Gomma area.

## 3. Materials and Methods

### 3.1. Study Area: Gomma District, Jimma Zone

The current study was conducted in Gomma District, found in Jimma Zone, Oromia National Regional state, Ethiopia. The district is located 397 km southwest of Addis Ababa and about 50 km west of Jimma town, capital of the Zone (Figure 2). Its area is about 1230 km². The Gomma Oromo society is largely Muslim, speaks native Afan Oromo, and has lived in the area for centuries. The community population is estimated to be over 200,000, although latest census figures are not known.

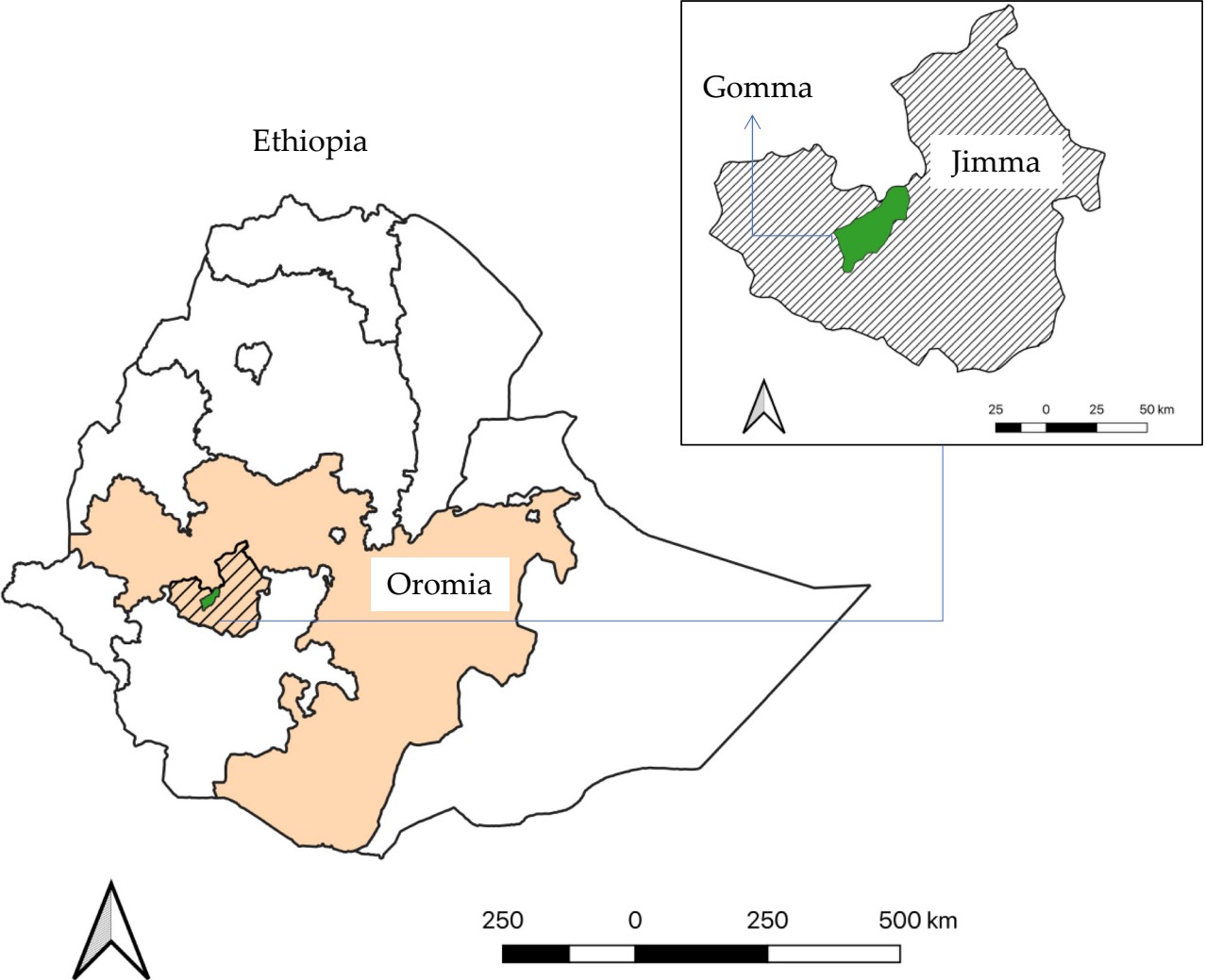

**Figure 2.** The location of the study area (Gomma district) in Oromia Regional state, Ethiopia.

The annual rainfall of the area varies between 1200 and 1800 mm [25], while the mean minimum and maximum annual temperatures vary between 7 and 12 °C and 25 and 30 °C [26]. The community is largely agrarian with rainfed agriculture as its main livelihood activity. Major crops grown include coffee (*Coffea arabica*), maize (*Zea mays*), teff (*Eragrostis tef*), sorghum (*Sorghum bicolor*), and enset (*Ensete ventricosum*). Coffee is the main tree commodity in the area and the basis of the economy: coffee is the main cash crop and source of income. According to Petty et al. [25], almost one-third (32.7%) of the households' income comes from coffee and 14.2% comes from coffee-based employment. Hence, coffee generates almost half of the household income at any point in time.

The Gomma district community claims that coffee's original identification took place in their own area. Specifically, coffee is said to have been discovered around the 10th

century by a herder named Khalid who suddenly noticed one day that his goats were behaving erratically and energetically (elders of the community and discussants of this study claim that the name of the herder is often incorrect and that he was indeed Khalid not Khaldi or Kaldi as it is widely known in most historical accounts of coffee identification):

*"Coffee was first identified in Gomma district at Kattaa Muuduu-Gahaa in Coocee around 10th Century by Khalid who was shepherd of goats. Once up on a time, Khalid realized that his herd of goats started behaving differently and dancing after eating the leaf of a tree in the forest. Then, Khalid started investigating the leaf of a tree with suspicion and query. Unfortunately, there were merchants who came from different countries and crossed Gomma on their journey. These merchants were from Maji, Wollo and Yemen and one day they reached at Khalid's home during nighttime and asked him to let them sleep overnight at his home. Khalid welcomed them to his home. During nighttime, the guests observed the goats behaving abnormally and asked Khalid what the goats were eating. He told them what he had observed. The guests requested him to take them to the forest with herd of goats early in the morning to identify the tree and see the activities of the herd of goats. In the morning, he took them to the area. When they reached at the place, the goats immediately ran in to the forest to eat the coffee leaf".* A translated personal account of an interviewee, age 80.

Another key informant, male and aged 70, described in the Afan Oromo language how Khalid and the guests identified the coffee shrubs in the forests. The following is a translation:

*"Early in the morning Khalid took the guests and moved into the forest to look after the goats where they found the goats eating from a shrub with red cherry in the forest. They also found the birds were eating the red cherry. Khalid and the merchants then collected the red coffee cherry and the green coffee bean in two different bags. When they reached home, they prepared a hole with fire and added the coffee beans into the hole. After burning, the coffee bean smelled good.. The next day they roasted the beans on a clay pan and crushed them with a stone mortar to eat as porridge. They ate it with wooden spoons and felt the stimulant nature of the coffee. Seeing this, the merchants took the coffee bean in their bags and spread it around while going through the country during their long-distance travel for trade. The local Oromo community around Gomma district noticed that the forest of Kattaa Muuduu-Gahaa had coffee plants, as noticed by Khalid. The local society started to protect the forest and the coffee shrubs in the forest for wider use".*

Study participants confirmed that after coffee was noticed by Khalid around Kattaa Muuduu-Gahaa, the local Oromo community started to conserve coffee plants in the forest and even planted coffee shrubs around their homesteads and farms. To this day, most of the coffee in this area is grown as forest coffee with a small share grown in home gardens. The inhabitants of this area know this resource to be of immense global importance, as also indicated by Hein and Gatzweiler [27].

### 3.2. Data Collection Methods

The study relied on qualitative data collected using in-depth interviews, focus group discussions, and key informant interviews. The use of these three methods of data collection helps to ensure high-quality data are collected, because the data are qualitative by their nature and need crosschecking to avoid personal biases from respondents/participants. A respondent only participated in a single data collection method. In total, about 59 respondents took part in the different data collection methods described below.

In-depth Interviews: In-depth interviews were employed to collect data from interviewees based on interview guidelines (see annex). The researchers used both structured and unstructured interviews. Interviews were conducted with coffee farmers, coffee makers (women who brew coffee and sell it in settlements), and elders. In-depth interviews were conducted with twenty individuals. See Table 1 for the participant types and details.

**Table 1.** Methods of data collection and types of respondents involved.

| Method | Respondents or Participants Types | Number of Respondents |
|---|---|---|
| In-depth interviews | Farmers (coffee producers) | 10 |
| | Coffee makers and sellers | 7 |
| | Community elders | 3 |
| Key informant' interviews | Farmers (coffee producers) | 8 |
| | Public servants working at Gomma District Cultural and Tourism Bureau | 4 |
| | Community elders | 3 |
| Focus group discussion | Farmers (coffee producers) | 8 |
| | Coffee makers at home | 5 |
| | Public servants from Gomma District Cultural and Tourism Bureau | 7 |
| | Elders from local communities | 4 |

Key Informant Interviews: Key informant interviews were undertaken to collect relevant data from members of the community. These included government officials who work in offices that promote coffee-related culture and values, such as the Cultural and Tourism Bureau. See Table 1 for the participant types and details.

Focus Group Discussions (FGDs): Focus group discussions were carried out with groups of eight individuals to discuss the value of coffee in their household and community. In total, three FGDs were conducted, making the total number of participants 24 (see Table 1 for the participant types and details). Each FGD lasted for about 2 h.

### 3.3. Data Interpretation and Presentation

The collected data were transcribed and converted into structured text that served as the database from which details were extracted and presented with further elaboration. This was largely an ethnographic study; therefore, the presentation of the results is based on the contextual details provided. See Appendix A for data collection methods.

### 4. Results Findings and Reflections

### 4.1. Coffee's Symbolism and Utilities in the Context of the Gomma Community

Coffee in Gomma community is playing a much broader role than what most coffee users globally see and know. Figure 3 summarizes the broader symbolic and utility uses based on the respondents' accounts.

Coffee serves both as a utility and as symbolic representation to the Gomma community (Figure 3). The utility functions are the commodity (market item) nature of coffee, coffee foods, and coffee as livelihood object. Most people are aware of such roles played by coffee in fulfilling specific societal needs or wants. The habitat roles played by coffee and the associated forests are less well known, but also are also sometimes defined as a utility. The roles that coffee play in society in the form of social instruments, cultural and religious functions, and communication media are generally categorized as symbolic functions. The reason for this is that coffee in such roles symbolizes some unique meanings or contexts which are often specific to a given society, culture or environment. A more elaborate discussion on each of the utility aspects and the symbolic nature of coffee appears in the subsequent sections.

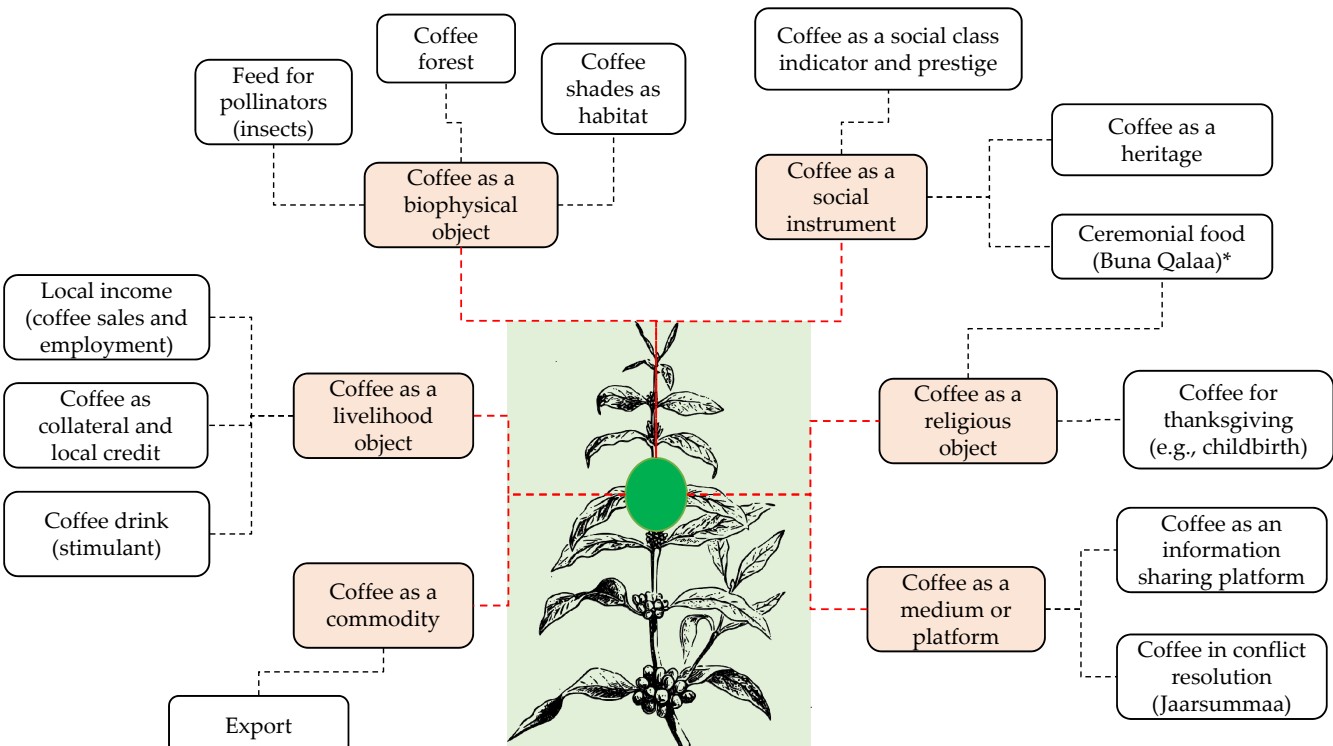

**Figure 3.** Summary of the symbolic and utility roles of coffee among communities in Gomma area. * Note that "Buna Qalaa" is used during special events and is not a mainstream food.

### 4.1.1. Coffee as an Economic Pillar

Income and employment: Coffee is the main cash crop grown in this community. The sale of coffee produced in the area covers close to one-third of the income of households (32.7%) [25]. The community members interviewed expressed this in their saying, "*Bunni keenya waan hundumaa keenya*", which literally means, "Our coffee is everything for us". This sentiment reflects the very strong dependency of the community on coffee. The discussants, in one voice, expressed how the community could not exist without coffee by saying "*Uummatni Gommaa jiruu fi jireenyi isaa bunaan kan gadi dhaabbatedha*", which literally is saying that "the lives and livelihoods of Gomma people are strongly dependent on coffee". This is also because, as Petty et al. [25] revealed, coffee-based employment generates approximately around 14.2% of the household income source in this district. Women in the area have also begun making coffee that is sold on the roadsides to make small earnings. Although the majority of residents are not in any formal employment, coffee generates what few employment opportunities are available in the area. The fact that the area is also believed to be the origin of coffee attracts tourists and thereby generates some job opportunities.

It is further important to note that a significant share of the coffee is produced under forest conditions and that the forests are an important source of other benefits that support the community. The discussants confirmed that forest trees serve as a source of firewood and construction wood, and that forested areas were important for hunting, bee-keeping, and other livelihood activities. This emphasis by the communities echoes the reported benefits of forests in this area by Tadesse et al. [11], the authors of which found that provisioning services from the forests account for about 30–75% of the total household income. The study reported that the value of sales of all coffee forest-based services per household per year was about USD 827, and that the direct market income from provisioning services from forests and coffee agroforests was USD 570 per hectare per year.

Coffee-based food and drink: Among the Gomma society and the wider Oromo community, there is a traditional edible item called "Buna Qalaa", prepared from freshly

collected coffee beans boiled and roasted in butter. This food is so prestigious that it is only prepared on unique occasions like weddings, religious ceremonies, or celebrations. Various types of edible foods and drinks prepared from coffee such as coffee roll (mix of ground coffee, butter and meat) and "Qorii" (roasted coffee and wheat mixed in melted butter) were also reported by Senbeta et al. [28] and Duressa [29]. The most widely popular form of coffee use is the coffee drink. Among the Gomma society, coffee is typically drunk three times a day—morning coffee (*Buna ganamaa*), midday coffee (*Buna guyyaa*), and evening coffee (*Buna galgalaa*). All three coffee drinking times have their own societal meanings and purpose. Morning coffee is the coffee session of the family where prayers for the day are held, thanksgiving for the past night is made, and directions for the activities of the day are given by the head of the family or their representative. Every member of the household will learn of their tasks during the morning coffee session. The midday coffee (usually around 1:00 p.m.) is to energize the family members as they come from farm activities for a break. Discussions about field activities will only happen if there are serious issues. Evening coffee is as important as morning coffee, because at this coffee session reflections on how the day went are given by every person assigned to a task. This is when challenges and successes are discussed at a family level. There is also a common tradition of inviting neighbors for coffee in the morning and evening coffee. When neighbors are present, the discussions in the coffee session focus on broader community issues. Hence, coffee as a drink plays a very crucial role in family and community interactions. Such household and neighborhood level meetings are unlikely to take place without coffee. Previous studies such as Verma [30] elaborated on the ceremonial aspects of coffee drinking events.

### 4.1.2. Coffee as a Local (Traditional) Medicine

According to the key informants and respondents of the focus group discussions, coffee in various forms is a cure for wounds, diarrhea, headache, and pain. To treat wounds, ground coffee is mixed with honey and smeared on the wound. This same mixture taken orally is also used to treat diarrhea. For headaches, strong coffee is usually prepared; once the person drinks it, there is usually a major relief of pain.

Coffee also plays a key role in destressing. Often, people come to the coffee ceremonies and discuss their issues, with elders of the community advising on what to do in the cases of specific individual problems. The elders' solution or community deliberation of one's personal problem usually leads to attenuation of the problem, either via experiences of the discussants or solutions proposed by the community.

### 4.1.3. Coffee as a Key Element of Celebration and Thanksgiving

Coffee is a key ingredient of numerous psycho-social support activities in the community, such as bereavement (mourning), childbirth, calf birth, burial, and welcoming of guests. In Gomma culture, it is a tradition to take coffee (beans, roasted, or boiled) to the grieving family to symbolize standing with them in such a difficult time. This coffee is usually accompanied by food items. However, the discussants stressed that the most important aspect is the coffee, with the community less concerned about the food gifts. The main reason why the community emphasizes the coffee here is because for a grieved family, the main source of encouragement is by staying with them and talking to them so that they gradually forget the loss. Therefore, during these long hours of chatting and discussion, coffee drink is prepared and provided to the numerous visiting guests. It is impolite to treat visiting guests without providing coffee. A common saying strongly conveys the central role of coffee during mourning: "*Namni mana boo'ichaatti buna hin geessine, ofumaayyuu du'eera*" translates as "*A person who does not take coffee to the grieving family is dead himself/herself*'".

Coffee is also a key ingredient for celebration during childbirth and calf birth. There are four distinct stages in which coffee plays a crucial role in the process of childbirth:

- As soon as the expectant mother goes into labor, elders are called for a special prayer for the mother. As soon as they arrive, coffee is roasted and one of the elders takes

the roasted coffee and prays for the woman. He says "*Ani siif hiikkadheeraa, Rabbi siif haa hiikkatu*" ("*As an elder I forgive you ('release' you from the pain of delivery), Let God the Almighty also relieve you from the pain*"). The belief is that coffee is a gift from God; it is used as a communication medium between humans and the creator. Further coffee is roasted and used for prayer after the expectant mother delivers;

- Once the mother delivers, coffee is again crucial when community members are invited for celebration. The invitation is phrased "*Deessuun keenya hiikamtee, kottaa buna dhugaa Waaqa galateeffannaa*", which roughly translates as "*Our expectant mother has safely delivered, so let us have coffee and celebrate*". Coffee has a central role in thanksgiving;

- The third stage is the collective celebration of the baby's birth. A special thanksgiving ceremony called "*Buna Ofkaltii Maaree*" (coffee for thanksgiving ceremony for safe delivery) is held. All visitors to the mother say "*Ilmoon kee siif haa guddattu, ilmoon ilmoo leencaa haa taatu*" ("*May your child grow like a lion cub*"), wishing the baby to grow to be brave and strong. They also add the statement "*Siree cabsii ka'i*" to the mother, meaning "*May you get strong quickly and get off the bed*", a recognition of the strain of childbirth on the mother. Coffee for this celebration is prepared following strict principles. If the newborn is a boy, the woman who prepares the coffee stands behind the coffee pot, opens and closes the pot five times, and five rounds of ululation are made. If the baby is a girl, the number of times the pot is opened is four, and so are the ululations. The reason for one fewer opening of coffee pot and ululation for the baby girl is that by tradition, she will be married when she grows up and will obtain a share from her husband's family. The one missing ululation also is seen as an affirmation they are praying that indeed the girl grows, marries, and receives her share from the other side. The community does not see it as discrimination against females;

- The fourth stage is what is referred to as "*Sirna Shananii*", the collective celebration ceremony on the fifth day after delivery. This is the day that the mother who gave birth prepares to be active again after a few days of rest. During the thanksgiving ceremony, roasted coffee is prepared and is held as the prayer goes on. In this ceremony, coffee serves three purposes: as a prayer medium, as food (*Buna Qalaa*), and as a normal coffee drink for participants. There is a general belief that God hears their thanksgiving using coffee as a medium of connection, because God gave them the plant as a blessing.

Coffee is also a key ingredient during celebrations of cows producing a calf. When a cow gives birth, a similar celebration takes place accompanied by coffee food (*Buna Qalaa*) and coffee drink. Among the Oromo community, cattle have a special place and are treated with the highest respect. This is why the Gomma Oromos, like all other Oromo communities, celebrate the birth of a calf. As soon as the calf is born, the Oromo tradition is that they collect a little milk of the cow daily for a month, leaving most of the milk for the calf. The little they take from the cow is accumulated separately, and one month after the calf is born, a special celebration is held. At this event, the ceremony can only be opened with a uniquely prepared coffee drink. Then, the accumulated milk is made into a food called "*silga*" that is served with "*Buna Qalaa*". This ceremony, just like the one for the baby birth, is a thanksgiving. The elders and the community bless and pray for the owners, and also pray that the cow and the calf like each other. This is important because the community believes that if there is a curse, the cow may reject giving milk to her calf and, although it rarely happens, the calf may ignore its mother. They believe that prayer breaks this curse.

Coffee is also used as a symbolic item for the expression of love and respect and as a welcoming gesture. If the new guest or visitor arrives and stays only for a short period, they are given coffee beans. If a new person or family arrives and settles in the area, the community elders provide coffee plants and land to grow additional coffee shrubs. Just like other cultures where coffee is common, any guest arriving is welcomed with the gesture of preparing coffee for them. Women of the household often invite guests by saying "*Nooraa bunan isiniif dhaabaa*" ("*Welcome. Get inside. Let me make coffee for you*").

### 4.1.4. Coffee as a Platform for Social Interaction and Conflict Resolution

Among the Gomma community, the process of conflict resolution starts with coffee first. A typical phrase among the Gomma and also among the wider Oromo community is "*Bunaa fi nagaa hin dhabinaa*" ("*May coffee and peace be with you all the time*"). The community equates coffee to peace, without which nothing can succeed. On some occasions, as soon as conflict resolution discussions are over, coffee has to be served. In other cases, talks to resolve conflict only start while or after drinking coffee. The coffee and its ceremony then serve as the platform for social interaction, which then facilitates dialogue and problem solving. There are emerging studies on social interactions in coffee shops in the developed world, too [31–33]. The studies all point to coffee increasing social interactions.

In the wider Oromo community, conflict resolution happens in a highly structured manner called "*Jaarsummaa*", which is a forum organized by community elders where cases are heard to amicably solve the problem in a fair and just manner. It can take hours to days, depending on the context of the conflict. At the start of this process, irrespective of the complexity of the situation, elders begin by praying to be guided by wisdom and to make peace without affecting any of the parties unfairly. People with cases are invited to come and sit in front of where women make coffee, and the complainants are the ones to drink first. Case hearings can only start once coffee is drunk. When the discussants were asked what the role of coffee in this process is, they responded by saying "*lafa bunni jiru nagaan jira*" ("*where there is coffee, there is peace*"), the implication being that at the coffee ceremony, one can discuss issues and resolve matters peacefully. The sharing of coffee from the same pot also signifies the sprit and desire for reconciliation. The discussants said "*Kan waliin nyaate wal hin nyaatu*" ("*If you share a meal or coffee together, you cannot dare to be enemies anymore*"). In Gomma, and among the wider Oromo people, the belief is that there is no local issue that cannot be resolved by elders. The respect for elders is almost total. Once the peacemaking process is complete, and the issues have been deliberated, verdicts are given that are always upheld by both complainants.

Coffee ceremonies, either at a family or communal level, serve as the main platforms for information exchange as well. Through the chats and discussions happening at this ceremony, people learn know what is going on in their localities, whether it concerns development, a disease outbreak, intruders, or even the state of natural resources in their areas. It is therefore a crucial communication medium in such remote locations where modern communication facilities are not easily accessible.

### 4.1.5. Coffee as Social Credit Facility and Collateral

There is limited access to credit facilities and other sourcing of financing when needed; therefore, Gomma communities use coffee stands as a collateral when borrowing money from each other through a social credit—a credit scheme whereby the members of community help each other in difficult times in anticipation of repayment when coffee is mature in the year. Any failure to repay the loan may thus lead to a likely loss of coffee stands based on the value estimated by local elders. Moreover, one can also borrow money from other members of the community on the anticipated yields of the coffee stand that they own. The repayment of the loan interest occurs when the coffee berries are collected. Such a practice is often called "*Araaxa*" among the Gomma community. Overall, if one has large coffee stand, the likelihood of accessing locally available lending sources becomes easier. That is why the community says about coffee, "*Bunni keenya waan hundumaa keenya*". This literally means, "*Our coffee is everything for us*".

### 4.1.6. Coffee Defining Social Class and Prosperity

The Gomma community also has its own social classes. This social class or structure is largely based on wealth and family history. Wealth is mostly defined by the size of coffee farms, livestock numbers, and agricultural farm sizes. A family with a large coffee farm, many crop fields, and a large number of livestock is often considered rich. The rich usually harvest large quantities of coffee from their relatively extensive coffee stands. One

can move up the social class by also buying coffee farms; the farms can easily be sold locally. Sometimes, access to certain social positions (e.g., village elders, elected positions) is influenced by the size of a person's coffee farms because, by and large, the richer the family, the more respected they are.

### 4.1.7. Coffee as a Catalyst for Ecosystem Conservation

The presence of coffee in the surrounding forests has played a major role in conserving the forest vegetation. Without shade trees in the forest ecosystems, coffee production would not be possible. The economic and cultural values of the crop, according to the key informants and discussants, helps to avoid forest loss. This, however, does not mean that forest is loss is not happening. Significant loss is occurring, especially in communities that do not understand the in-depth connection between the coffee forests and the existence and livelihood of the Gomma. The community even prides itself on being the widely known origin of coffee. Asked about the "what if there is no coffee in the forest" scenario, many in the community admitted that without coffee, the surrounding forest would have been lost already. The coffee forests are therefore, by extension, refugia for myriad other forest species that could have already been lost from the area. Over 60% of the species that occurred in native forest fragments also occurred in semi-forest coffee stands, as reported by Tadesse et al. [34]. This concurs with the sentiments expressed by the local communities that coffee is playing a role in reducing the loss of biodiversity. Other studies have also emphasized the key role that coffee forests play in maintaining vegetation diversity [9,35]. As with the communities, Tadesse et al. [36] argued that any replacement of coffee forests with even a heavy shade coffee agroforestry could affect the coffee forest-based cultural, regulating, and supporting services because these services are largely concentrated in the forest remnants that are native habitats for wild coffee.

It is finally important to note that the communities' forest and semi-forest coffee production systems do not use chemicals intensively, and hence are friendly to pollinators, insects and other wild animals; they are nature-friendly production schemes.

### 4.1.8. Coffee as an Inheritance and Heritage

Among the Gomma community, coffee is one of the most cherished inheritances that can be passed to the next generation. Parents who manage to hand down coffee stands to their children take great pride in the fact, knowing that with proper farm management (e.g., weeding, pruning, thinning, tree shades), coffee bushes never stop yielding and that their children will benefit. There is high regard for a coffee stand that has passed on from generation to generation, and that offspring will manage the farm in memory of their parents, while on the side they also develop their own plots to pass on to their own children. Overall, coffee stands are seen as intergenerational heritage which passes on with proper management. The use of trees as collective heritages is also reported in other countries [37–40].

## 5. How to Save the Coffee Forests of Southwest Ethiopia?

For communities such as those living in Gomma, their entire life rotates around coffee. Therefore, the ongoing impact of human activities and climate change on coffee, as well as the threat of further climate change and variability impact, does not just concern the loss of livelihoods, but the loss of a whole way of life. Thus, interventions to help coffee systems adapt are of paramount importance. This adaptation need is not just because the communities depend on this commodity crop, but also because the wild genetic pool of coffee might vanish if no action is taken. Hence, the call for measures to conserve such economically, socially, environmentally, and genetically important forests has to be louder than ever before, so that deliberate actions are taken to save the forest and the livelihoods and ways of life that depend on it.



For immediate actions that could be taken to save this precious national and global resource, incentives that support the local communities to continue conserving this forest are vital. Incentives could be generated in numerous ways:

- Promoting and rewarding organic coffee: forest and semi-forest coffee are among the most organic coffee production systems. Recognizing this and adding a price premium for producer communities based on their conservation efforts could be highly motivating. This should be supported by a clear offtake commitment by the collectors and processing companies, both domestic and foreign. Such long-term commitments should be enforceable based on the terms and conditions accompanying the agreements;

- Promoting coffee-based tourism: this could be "low hanging fruit" for action to save these important forests. With local and international non-governmental organizations, the national and regional governments should take action to promote coffee-based tourism. With such interventions, the level of awareness increases, and communities may earn additional livelihood benefits;

- Promoting coffee forest-based sustainable enterprises: among the alternative interventions to save the coffee forests is developing and investing in sustainable forest-based enterprises such as bee-keeping, spice production, and non-wood products for household furniture. However, for this to be effective, there is a need to create marketing mechanisms, technical support such as training, and financial support—particularly, startup resources should be availed;

- Carbon credit schemes: with REDD+ (reducing emissions from deforestation and forest degradation) in its implementation phase after the Paris Agreement, coffee forests could be a priority to achieve the emission reduction targets. In particular, they can be marketed at premium prices due to their genetic values too. However, the caveats of REDD+ emphasize benefit sharing and rewarding local communities rather than benefiting companies that sell the emission credits. A fair share of the benefit should go to the local communities. It is important to note that due to the current low carbon pricing level, REDD+ may not be a stand-alone solution but rather one that complements other livelihood-generating interventions;

- Payment for ecosystem services: the coffee forests are also sources of numerous ecosystem services that are crucial for the wider southwest part of Ethiopia. Incentives for communities could be created by designing locally appropriate payment for ecosystem services mechanisms. Such payment could even just be conserving the wild coffee gene pool which is estimated to be worth billions [27]. However, as with carbon mentioned above, the caveats of benefit sharing, and inclusive intervention designs, need to be addressed. This is highly important; ecosystem services are not an individual matter but rather a collective concern, because ecosystems span ownership boundaries. What may be required is a collective (communal) approach because the generation of ecosystem services (e.g., water, habitat) involves treatment of landscapes. This then may require collective rights and tenure arrangements with the help of respective government agencies.

## 6. Concluding Thoughts

This study aimed at exploring the unexplored socio-cultural benefits of coffee in the Gomma area with an ethnographic study approach that deployed symbolic interpretation and utility as the guiding frames. Gomma area, where the first coffee was discovered, is almost certainly the origin of the coffee gene pool. Despite such high global economic and biodiversity importance, little is known about the way coffee manifests in the daily life of the people of Gomma. About 59 people from the district participated in the data collection via key informant interviews, in-depth interviews, and focus group discussions. We observed a very strong connection between the community and coffee, well beyond the usually emphasized economic values of the crop. Coffee is part of every aspect of their life, from food to medicine, from being a medium for conflict resolutions to an ingredient for



blessing newcomers to the community. Symbolically, coffee represents much of what is prized in life—procreation, human relationships, peace, wealth, prestige, access to credit, having an asset to pass to descendants, a healthy, shaded, and well-watered environment, and much more.

Despite conferring these multiple benefits and services, however, coffee forests are facing threats from humankind through land use conversions, including investors promoting coffee plantations, as well as current and future threats from climate change and variability. Today, coffee forests in Jimma zone where Gomma is situated resemble islands surrounded by agricultural areas. The pressure from competing land uses is growing, and measures are needed to make coffee forests attractive to the communities to conserve them further. Some identified incentives for the community to take up the management and better care for the ecosystem include building a proper incentive packages through premium prices, carbon credits, payment for ecosystem services, and tourism, to the area which is the origin of arabica coffee in particular.

The results presented in this study point to the need for seeing coffee beyond how it has largely been viewed, so that it receives the attention it deserves. Coffee is far more than just a commodity crop from which communities generate livelihood. The realization and inclusion of these additional values builds a case for a stronger action by responsible agencies (both governmental and non-governmental). The coffee forest that is home to such precious global resources should be prioritized for conservation and wise use, in close collaboration with the local communities. The lack of an exhaustive synthesis of the benefits of the crop at local scale may have hindered the conservation emphasis placed on it. We believe such in-depth societal perspectives of crops that the planet is intimately familiar with are needed if we are to save them from unwise use and irreversible exploitation. Today, billions of people enjoy coffee every day. The more they and key actors know the value of coffee at local level, the more they will appreciate the worth of the crop at multiple scales—local, national, and global. The loss of coffee forests is not just a loss for the communities that rely directly on them, but a loss for the planet, which would lose a significant asset (gene pools) and the intangible benefits that the coffee trees and crop used to provide.

To strengthen efforts to conserve the crop and its forest, there is a need for further research on how much the values are worth so that a proper compensation for the resources is availed to the community.

**Author Contributions:** Conceptualization, B.J.B. and L.A.D.; methodology, formal analysis, B.J.B. and L.A.D.; writing—original draft preparation, B.J.B. and L.A.D.; writing—review and editing, B.J.B. and L.A.D. All authors have read and agreed to the published version of the manuscript.

**Funding:** The authors acknowledge the Forest, Trees and Agroforestry (FTA) Program of the CGIAR for supporting the APC.

**Institutional Review Board Statement:** Not Applicable.

**Informed Consent Statement:** Not Applicable.

**Data Availability Statement:** Data scripts for this research are available from the authors upon request.

**Acknowledgments:** We are very grateful for the study area maps produced by Getu Lami of Jimma University. We are grateful to Tesfaye Woldeyohanes (World Agroforestry—ICRAF) for comments on the initial draft of this manuscript. The Forests, Trees and Agroforestry (FTA) program of the CGIAR supported the staff time of L.D. while working on this document. We are grateful for language edits on the early drafts by Esther Kamwilu. We are very grateful for the professional language editing by Cathy Watson.

**Conflicts of Interest:** The authors declare no conflict of interest.

**Appendix A Guides Used for Data Collection**

**INTERVIEW GUIDES**

*Farmers (Coffee Producers)*

1. Why do you plant coffee around your homestead?
2. What challenges do you face to produce coffee?
3. How many quintals of coffee do you produce each year?
4. Do you grow coffee in the forest?
5. What economic benefits do you get from coffee production?
6. Is there any price fluctuation each year?

*Coffee makers at home (coffee ceremony)*

1. Why do you make coffee at your home?
2. How often do you make coffee per day?
3. What kinds of cultural food do you use for coffee ceremony?
4. What kinds of food do you make from coffee?
5. For what purposes do you use coffee?
6. What kinds of issues do you discuss during coffee ceremony?
7. During what occasions do you prepare coffee ceremony?

*Coffee makers (selling coffee) for business*

1. When did you start to make coffee for selling?
2. What pushes you to make coffee for your business?
3. What are the economic benefits you earn from making coffee for drinking?
4. What challenges do you face to make coffee for your business?
5. From whom do you get coffee beans?
6. Do you have another source of income beside this?

**GUIDES FOR FOCUS GROUP DISCUSSION**

*Farmers (Coffee Producers)*

1. Discuss the economic benefits of coffee for your life.
2. Explain the ecological values of coffee plantation.
3. Discuss factors affecting coffee production and distribution.
4. Describe why you plant coffee around your homestead.
5. Explain where and when you plant coffee.

*Coffee Makers at home (coffee ceremony)*

1. Discuss how often you make coffee per day at your home.
2. Explain the issues you discuss during coffee ceremony.
3. Discuss the kinds of traditional food you make from coffee.
4. Describe the social uses of coffee ceremony.
5. Discuss the uses of coffee for your social and cultural lives.

*Coffee Makers for income generation*

1. Discuss the economic uses of making coffee for drinking as your small business.
2. Describe why and how you did start to do this business.
3. Discuss the benefits you earn from coffee production.
4. Explain why you started to make coffee as a business.

*Coffee distributors*

1. Discuss different challenges that you face during coffee distribution.
2. Describe the economic values of coffee.
3. Explain where you get coffee to distribute.
4. Explain the contributions of coffee in your livelihood.

**KEY INFORMANT INTERVIEW**

*Elders of the community*

1. What are the traditional foods prepared from coffee?
2. Is coffee used for social gatherings and conflict resolution?
3. What are the social values of coffee?
4. What are the medical uses of coffee among the society of Gomma district?

*Officers of Culture and Tourism Bureau*

1. What are the sociocultural uses of coffee among the Gomma Oromo?
2. Is coffee used for the preparation of traditional food?
3. What kinds of traditional foods do the people prepare from coffee?
4. Does coffee symbolize the social life of the Gomma Oromo?
5. What are the symbolisms of coffee among the Gomma Oromo?
6. What are the cultural meanings of coffee consumption and production?

*Trade and Market Development Office*

1. What are the economic benefits of coffee for Gomma Oromo society?
2. How many merchants are dependent on coffee for their livelihood?
3. How many quintals of coffee are exported from Gomma district to the National market?
4. How many quintals of coffee are exported from Gomma district to the international market?
5. What factors are hindering the coffee trade in Gomma district?

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
