# Peer review of "The Unexplored Socio-Cultural Benefits of Coffee Plants: Implications for the Sustainable Management of Ethiopia’s Coffee Forests"

_sustainability, doi:10.3390/su13073912_

Round 1
Reviewer 1 Report
Thank you for submitting this interesting work to Sustainability. This is an interesting and timely study. Below are a few comments that could further improve the quality of this work:
-check your reference software, that is all over the place
-Fig. 2 maybe also an overview of Ehtiopia so that we know where the gomma district is
-What where the interviews about, did you develop a questionnaire? How did you make sure the results can be compared
-list participants in a figure to provide a good overview
-also mention the number of participants in the abstract and conclusions
Author Response
Comments and Suggestions for Authors
Thank you for submitting this interesting work to Sustainability. This is an interesting and timely study.
Below are a few comments that could further improve the quality of this work:
Reviewer Comment:: check your reference software, that is all over the place
Response: Thank you for the comments. We have corrected all the references in the revised version submitted.
Reviewer Comment:: Fig. 2 maybe also an overview of Ehtiopia so that we know where the gomma district is
Response: We think the Figure 2 addresses the comment. It has Ethiopia, in it the green shaded one is Oromia Region, in it is then Jimma zone, within which we showed the Gomma district.
Reviewer Comment:: What where the interviews about, did you develop a questionnaire? How did you make sure the results can be compared
Response: In this case, broadly, the idea is not comparing between the various groups but rather getting a coherent explanation of the various groups of people such as elders, coffee makers, the district board etc. whom we interviewed.
Reviewer Comment:: list participants in a figure to provide a good overview
Response: Thank you for the comments. We inserted a new table (Table 1) that presents the information listed.
Reviewer Comment:: also mention the number of participants in the abstract and conclusions
Response: A sentence capturing the requested detail was added in the concluding thought section.
Reviewer 2 Report
This paper reads well and is well organised and presented in the correct manner.
For those who know the area, as I do, there is little new in the paper but it is a good compilation and linkage to specific local understanding.
There are some gaps when considering sustainable management which might have been covered. These include:
- The challenges of sustaining the forest canopy when coffee is grown, so that the statements about maintaining coffee forest needs some qualification and references to the work of people like Dr Kitessa.
- The importance of clear rights to forest land and the role of participatory forest management in helping achieve active community management of the forest to prevent conversion to other uses.
- The need to explore a full range of forest-based enterprises to support wider economic development and interest in maintaining all forest, so that coffee forest does not become islands in a sea of agriculture as is the case in many areas near to Jimma.
Author Response
Comments and Suggestions for Authors
Reviewer Comment: This paper reads well and is well organised and presented in the correct manner.
Response: Thank you for recognizing the importance of the manuscript.
Reviewer Comment: For those who know the area, as I do, there is little new in the paper but it is a good compilation and linkage to specific local understanding.
Response: Thank you. In fact, the paper brings the less emphasized aspect of coffee growing systems which to date has not been systematically studied and presented in a structured manner. This did not exist to-date linking it with the community who claims they have the natural gene pool of coffee arabica.
Reviewer Comment: There are some gaps when considering sustainable management which might have been covered. These include:
- The challenges of sustaining the forest canopy when coffee is grown, so that the statements about maintaining coffee forest needs some qualification and references to the work of people like Dr Kitessa.
- The importance of clear rights to forest land and the role of participatory forest management in helping achieve active community management of the forest to prevent conversion to other uses.
- The need to explore a full range of forest-based enterprises to support wider economic development and interest in maintaining all forest, so that coffee forest does not become islands in a sea of agriculture as is the case in many areas near to Jimma.
Response: Thank you for the comments, we have picked the ideas you flagged and included it in the new Way Forward section.
Reviewer 3 Report
This was a very interesting study that will give many readers a better appreciation of the importance of coffee in its region of origin and some of the concerns regarding sustainability of coffee-producing forests in that region. The manuscript blurs the line between a paper that is more anthropological in nature and one that is more focused on sustainability, but it does this well.
There is some room for more description of the methods. In particular, how were participants selected for interviews and what efforts were made to select a representative group? What is the difference between an in-depth interview and a key informant interview? Did any individuals who were interviewed separately also participate in a focus group? What is the demographic make-up of those interviewed (male-female, age, level of education, etc.)? It may not be possible to cover all of this, but right now the descriptions are a bit sparse in terms of what is normally presented in the methods sections of papers using social science research methods.
The sentence on Lines 244-245 seems over stated. In an area with this much population there must be at least some employment opportunities other than those directly related to coffee.
In Line 263 I was curious what was meant by protein.
In Line 288 I'm guessing "string" should be "strong."
Section 4.1.7 (plus the "good management" reference in Section 4.1.8) made me wonder about the level of management of coffee stands (planting, tending, pruning, etc.) and also whether there is any active selection and breeding based on individual tree characteristics. I wouldn't say this is mandatory to include, but it could be of interest to many readers.
Author Response
Comments and Suggestions for Authors
This was a very interesting study that will give many readers a better appreciation of the importance of coffee in its region of origin and some of the concerns regarding sustainability of coffee-producing forests in that region. The manuscript blurs the line between a paper that is more anthropological in nature and one that is more focused on sustainability, but it does this well.
Response: Thank you so much.
Reviewer Comment: There is some room for more description of the methods. In particular, how were participants selected for interviews and what efforts were made to select a representative group? What is the difference between an in-depth interview and a key informant interview? Did any individuals who were interviewed separately also participate in a focus group? What is the demographic make-up of those interviewed (male-female, age, level of education, etc.)? It may not be possible to cover all of this, but right now the descriptions are a bit sparse in terms of what is normally presented in the methods sections of papers using social science research methods.
Response: We have worked on the methods section especially on the participants of the different methods to make much clearer. See the new table added (Table 1). Key informants interview is largely to shade light the broader issues to be discussed while in-depth interviews dig deeper into the identified issues by following lead statements either the interviewer prepared or raised by the respondents. Coffee makers are almost totally women because culturally men are not the best to do it proper as per the local tradition. When it comes to coffee farming it is a mix. Elders are mostly men. Our selection of respondents is guided by this local context.
Reviewer Comment: The sentence on Lines 244-245 seems over stated. In an area with this much population there must be at least some employment opportunities other than those directly related to coffee.
Response: Yes, though our statement is true, we have edited the sentence to reflect the key role coffee has in generating employment opportunities more than all other activities.
Reviewer Comment: In Line 263 I was curious what was meant by protein.
Response: corrected as meat. Protein is too general rightly.
Reviewer Comment: In Line 288 I'm guessing "string" should be "strong."
Response: corrected. Thank you.
Reviewer Comment: Section 4.1.7 (plus the "good management" reference in Section 4.1.8) made me wonder about the level of management of coffee stands (planting, tending, pruning, etc.) and also whether there is any active selection and breeding based on individual tree characteristics. I wouldn't say this is mandatory to include, but it could be of interest to many readers.
Response: We have elaborated on this in section 4.1.8. Rather it is better to use ‘proper farm management’ than generally saying ‘good management’. We provided also details of the management practices. Thank you.
Reviewer 4 Report
This is an interesting study and I really enjoyed reading the paper. It is nice written and you did a good job of exploring on the social-cultural values of coffee in Ethiopia.
I would first recommend the authors to check again the reference links and numbers, for example on line 37, Kufa should be [6] not [1]. Secondly, a bit more review of literatures on social values of coffee would be good to improve the introduction. Thirdly, elaboration in the solutions to improve coffee forest management in section 4 would be helpful.
Author Response
Comments and Suggestions for Authors
This is an interesting study and I really enjoyed reading the paper. It is nice written and you did a good job of exploring on the social-cultural values of coffee in Ethiopia.
Response: Thank you.
Reviewer Comment: I would first recommend the authors to check again the reference links and numbers, for example on line 37, Kufa should be [6] not [1].
Response: Thank you. We fixed all the references. We have used the software to adjust all the references.
Reviewer Comment: Secondly, a bit more review of literatures on social values of coffee would be good to improve the introduction.
Response: We have added more literature on the social aspects of coffee.
Reviewer Comment: Thirdly, elaboration in the solutions to improve coffee forest management in section 4 would be helpful.
Response: Thank you for the suggestions. We have provided further details in relation to collective management of the forest resources in the area. The way forward section is revised and enriched.
Reviewer 5 Report
Thank you for an engaging read. I learned a lot about the coffee-related food culture in Gomma. The topic is clearly relevant. Although I like the paper very much, I think it can be improved.
The major single problem of the paper is that the theoretical framework (2.) does not reflect properly in the results section. You need to take care to address the implications raised by the framework, and discuss them more thoroughly. Also, the research question needs to be spelled out more clearly, and the way it refers to the overall coffee literature. How does the paper contribute to the literature?
Your use of "triangulation" in the method section could be questioned, as many scholars would understand as triangulation a joint application of quantitative and qualitative methods and not different qualitative methods. At the same time, you do not need to mention that you did not use statistical methods. My recommendation is that you present this more self-confidently as an ethnographic study, and cut references to "triangulation" or "statistics".
Citation at one time is not consistent (line 389-390); you need to insert the numbers only.
Some general statements need to be referenced: "As in any modern society there is a class structure which is true in the Gomma community too" (lines 434-435); "Just like other cultures where coffee is common, any guest arriving is welcomed with the gesture of expressing to prepare coffee for them" (lines 374-375). You should be more specific here, which contexts do you mean? Which kind of knowledge are you refering to?
In my opinion, discussion of coffee as a commodity is underdeveloped in the text, and as a reader who likes anthropology to be connected to political economy, I would like to know more about this aspect. Who are the wholesalers? Is the coffee marketed globally ("single-plot origin")? Are there cooperatives? Fair-trade schemes? Are there volatile prices? How important are exports to livelihoods?
You refer to REDD+ – without further explaining the nature of the schemes – rather affirmatively assuming that this would benefit local communities. How do you account for the many negative and conflict-ridden aspects of REDD+ schemes (e.g. Chomba et al. 2016 – 10.1016/j.landusepol.2015.09.021; Myers et al. 2018 – 10.1016/j.gloenvcha.2018.02.015; Scheba & Rakotonarivo 2016 10.1016/j.landusepol.2016.06.028)?
Similar arguments could be made regarding payment for ecosystem services and organic certification. I find section 4 and 5 are the weakest parts of the paper. Half of section 4 touches on problems that should have been discussed much earlier, maybe in the introduction. The way they are presented now, they seem to be mere afterthoughts. The "concluding thoughts" are a mere summary and do not point to research gaps or limitations of the study. I would recommend to rewrite sections 4 and 5 and present a more efficient conclusion.
The language is good but at times not wholly accurate in its phrasing; I suggest editing by a native speaker of English, if possible.
Looking forward to reading an edited version.
Author Response
Comments and Suggestions for Authors
Thank you for an engaging read. I learned a lot about the coffee-related food culture in Gomma. The topic is clearly relevant. Although I like the paper very much, I think it can be improved.
Response: Thank you for your comments.
The major single problem of the paper is that the theoretical framework (2.) does not reflect properly in the results section. You need to take care to address the implications raised by the framework, and discuss them more thoroughly. Also, the research question needs to be spelled out more clearly, and the way it refers to the overall coffee literature. How does the paper contribute to the literature?
Response: There is scanty detail in the scientific literature on the values of coffee beyond its market aspects. This paper brings out the unexplored values and contributions of coffee to the social, cultural and religious practices of the community who believes coffee arabica was first identified in their area. The paper then elucidates the need to account for such wider values when looking at coffee and the forests that harbor the coffee plants. It is to challenge the general narrative that coffee is just a commodity and present with evidence on the wider benefits the plant provides.
Your use of "triangulation" in the method section could be questioned, as many scholars would understand as triangulation a joint application of quantitative and qualitative methods and not different qualitative methods. At the same time, you do not need to mention that you did not use statistical methods. My recommendation is that you present this more self-confidently as an ethnographic study, and cut references to "triangulation" or "statistics".
Response: Very true and we agree on the use of the ‘triangulation’ context. Also, We have removed the line that talks about statistical issues as there is no need for mentioning as you rightly pointed. Thank you.
Citation at one time is not consistent (line 389-390); you need to insert the numbers only.
Response: Thank you. We have fixed all the mentioned references.
Some general statements need to be referenced: "As in any modern society there is a class structure which is true in the Gomma community too" (lines 434-435); "Just like other cultures where coffee is common, any guest arriving is welcomed with the gesture of expressing to prepare coffee for them" (lines 374-375). You should be more specific here, which contexts do you mean? Which kind of knowledge are you refering to?
Response: Thank you. We have modified section 4.1.6 to address the above comments. All are addressed in greater details.
In my opinion, discussion of coffee as a commodity is underdeveloped in the text, and as a reader who likes anthropology to be connected to political economy, I would like to know more about this aspect. Who are the wholesalers? Is the coffee marketed globally ("single-plot origin")? Are there cooperatives? Fair-trade schemes? Are there volatile prices? How important are exports to livelihoods?
Response: We have captured much of this in the section on 4.1.1 Coffee as an economic pillar section. We have added details on coffee as a commodity to embellish the details raised. See the revised section.
You refer to REDD+ – without further explaining the nature of the schemes – rather affirmatively assuming that this would benefit local communities. How do you account for the many negative and conflict-ridden aspects of REDD+ schemes (e.g. Chomba et al. 2016 – 10.1016/j.landusepol.2015.09.021; Myers et al. 2018 – 10.1016/j.gloenvcha.2018.02.015; Scheba & Rakotonarivo 2016 10.1016/j.landusepol.2016.06.028)?
Response: Yes, such caveats are crucial to consider and we have reflected on those points in the Way forward section and on the possible safeguard mechanisms that need to be in
Similar arguments could be made regarding payment for ecosystem services and organic certification. I find section 4 and 5 are the weakest parts of the paper. Half of section 4 touches on problems that should have been discussed much earlier, maybe in the introduction. The way they are presented now, they seem to be mere afterthoughts. The "concluding thoughts" are a mere summary and do not point to research gaps or limitations of the study. I would recommend to rewrite sections 4 and 5 and present a more efficient conclusion.
Response: Thank you for the observations on the two sections and we have revised both immensely. Much of what is in the section ‘way forward’ has been moved into the introduction section to strengthen the problem framing. We suggest the editors and reviewers to see the revised introduction section, results section, Way forward section and concluding thoughts section.
The language is good but at times not wholly accurate in its phrasing; I suggest editing by a native speaker of English, if possible.
Response: Thank you. We in fact got the language edited by native speaker and former global media editor.
Round 2
Reviewer 1 Report
Excellent work
Reviewer 5 Report
Thank you, I find the manuscript much improved. However, there are still several language issues and I find editing was conducted maybe in a hurry, because there are many small typos that could be corrected easily. Please find some hints in the file attached, however, you should look for mistakes and grammar really thoroughly.
You need more citation for the methods section, and also some references for the section on REDD+
Best of success.
